# Isolation of Subtype 3c, 3e and 3f-Like Hepatitis E Virus Strains Stably Replicating to High Viral Loads in an Optimized Cell Culture System

**DOI:** 10.3390/v11060483

**Published:** 2019-05-28

**Authors:** Mathias Schemmerer, Reimar Johne, Monika Erl, Wolfgang Jilg, Jürgen J. Wenzel

**Affiliations:** 1Institute of Clinical Microbiology and Hygiene, University Medical Center Regensburg, 93053 Regensburg, Germany; monika.erl@ukr.de (M.E.); wolfgang.jilg@ukr.de (W.J.); juergen.wenzel@ukr.de (J.J.W.); 2Department of Biological Safety, German Federal Institute of Risk Assessment, 10589 Berlin, Germany; reimar.johne@bfr.bund.de

**Keywords:** hepatitis E virus, cell culture, whole genome, wild-type HEV isolation, high viral loads

## Abstract

The hepatitis E virus (HEV) is transmitted via the faecal–oral route in developing countries (genotypes 1 and 2) or through contaminated food and blood products worldwide (genotypes 3 and 4). In Europe, HEV subtypes 3c, 3e and 3f are predominant. HEV is the leading cause of acute hepatitis globally and immunocompromised patients are particularly at risk. Because of a lack of cell culture systems efficiently propagating wild-type viruses, research on HEV is mostly based on cell culture-adapted isolates carrying uncommon insertions in the hypervariable region (HVR). While optimizing the cell culture system using the cell culture-adapted HEV strain 47832c, we isolated three wild-type strains derived from clinical specimens representing the predominant spectrum of HEV in Europe. The novel isolates 14-16753 (3c), 14-22707 (3e) and 15-22016 (3f-like) replicate to high viral loads of 10^8^, 10^9^ and 10^6.5^ HEV RNA copies/mL at 14 days post-inoculation, respectively. In addition, they could be kept as persistently infected cell cultures with constant high viral loads (~10^9^ copies/mL) for more than a year. In contrast to the latest isolates 47832c, LBPR-0379 and Kernow-C1, the new isolates do not carry genome insertions in the HVR. Optimization of HEV cell culture identified amphotericin B, distinct salts and fetal calf serum (FCS) as important medium supplements. Overconfluent cell layers increased infectivity and virus production. PLC/PRF/5, HuH-7-Lunet BLR, A549 and HepG2/C3A supported replication with different efficiencies. The novel strains and optimized cell culture system may be useful for studies on the HEV life cycle, inactivation, specific drug and vaccine development.

## 1. Introduction

The hepatitis E virus (HEV) is a small, (+) single-stranded RNA virus belonging to the family *Hepeviridae*, genus *Orthohepevirus*, which is comprised of four species. *Orthohepevirus B* is found in chickens and *Orthohepevirus D* in bats [1]. *Orthohepevirus C* mainly circulates in rats but has zoonotic potential [2]. HEV of *Orthohepevirus A* is currently classified into seven genotypes (gt) [1,3] of which 1–4 [1] and 7 [4] infect humans. Genotypes 1 and 2 are restricted to humans and are transmitted via the faecal–oral route in developing countries. Genotypes 3 and 4 are also found in pig, wild boar and deer (amongst others) and are transmitted through contaminated food and blood products worldwide [5]. Genotype 7 is linked to consumption of contaminated camelid meat and milk [4]. Moreover, a putative genotype 8 was discovered which infects Bactrian camels [6]. The 7.2 kb long genome comprises three major open reading frames (ORF). ORF1 encodes the non-structural polyprotein and ORF2 for the capsid protein [7]. Of note, ORF2 is translated into different forms of capsid protein and only a minority is associated with viral particles whereas the free form is abundantly secreted [8,9]. ORF3 is important for viral release and encodes an ion channel [10]. An additional ORF4 is solely expressed by HEV gt 1 and controls the activity of the RNA dependent RNA polymerase [11].

The virus is spread worldwide whereas the most common subtypes in Europe are 3c, 3e and 3f [12]. The life cycle of the virus is not fully understood nor is the receptor described yet which is partly due to the fact that the virus cannot easily be cultivated [13]. The first successful approach generating high viral loads was the isolation of a subtype 3b strain in A549 and PLC/PRF/5 [14]. This success could be repeated with a subtype 4c strain [15] and was then confirmed by isolating 17 strains out of 23 HEV-positive sera [16]. Moreover, this cell culture system was also found to be suitable for isolating swine and wild boar HEV strains [17].

However, this approach could not be reproduced by several other workgroups [18,19,20] and since then, only four more strains could be isolated which replicated to high viral loads. Remarkably, three of these strains (47832c, LBPR-0379 and Kernow-C1) were isolated from patients with chronic hepatitis E and have insertions in the hypervariable region (HVR) of ORF1 [21,22,23]. These are associated with a growth advantage in vitro and are not commonly found in wild-type HEV. The insertions are thought to be acquired by recombination events either in the patient or in a previous host [21,23]. The fourth strain was derived from an experimentally infected pig and exclusively replicated in three-dimensional PLC/PRF/5 cell cultures generated in a rotating wall vessel [19]. Another approach is to use full-length infectious clones to transfect cells to generate infectious HEV [13]. This led to the development of several infectious HEV strains like G3-HEV83-2-27 [24], which was successfully used in research on the HEV life cycle [25].

Therefore, the currently available HEV cell culture systems are either not fully reproducible, require high maintenance effort, involve sophisticated equipment, rely on transfection or use cell culture-adapted strains carrying unusual genome insertions. It is debated whether these strains reflect the behavior of wild-type HEV [26]. The lack of a reproducible cell culture system susceptible to wild-type HEV also hampers the approach to generate a classic inactive or attenuated vaccine [27], to develop and test specific drugs or to identify strategies for efficient HEV inactivation [28].

This work aimed to improve and facilitate the HEV cell culture by (i) investigating if the high maintenance effort of daily medium refreshment can be reduced; (ii) analyzing the influence of cell confluence on susceptibility to HEV; (iii) testing several medium supplements; (iv) comparing the cell lines most permissive to HEV and the cell clone HuH-7-Lunet BLR which is highly permissive for hepatitis C virus RNA replication [29]; and (v) investigating the long-term stability of HEV producing cells. The optimized cell culture system was successfully used for virus isolation from human clinical specimens containing currently circulating HEV subtypes. Finally, the novel strains were further characterized.

## 2. Materials and Methods

### 2.1. Cell Culture

Lung carcinoma cell line A549 (ATCC CCL-185), liver carcinoma cell lines PLC/PRF/5 (ATCC CRL-8024) and HepG2/C3A (ATCC CRL-10741, all from LGC Standards, Wesel, Germany) as well as cell clones A549/D3 [30] and HuH-7-Lunet BLR (kindly provided by Prof. Dr. Ralf Bartenschlager, University of Heidelberg, Germany) [29] were maintained in BMEM (see Section 2.1.1 for detailed medium compositions) at 37 °C and 5% CO_2_.

#### 2.1.1. Cell Culture Media Composition

Cell culture reagents were purchased from PAN Biotech (Aidenbach, Germany), specific salts from Sigma-Aldrich (St. Louis, MO, USA).

BMEM:Eagle minimum essential medium (MEM) supplemented with 10% heat-inactivated fetal calf serum (FCS), 2 mM l-glutamine, 1% non-essential amino acids (NEAA), 100 U/mL penicillin G and 100 µg/mL streptomycin.MECK:BMEM additionally supplemented with 2.5 µg/mL amphotericin B, 10 mM CaCl_2_ and 10 mM K_2_SO_4_MEMM:BMEM additionally supplemented with 2.5 µg/mL amphotericin B and 30 mM MgCl_2_BMEM_G:Eagle MEM supplemented with 10% heat-inactivated FCS, 2 mM L-glutamine, 1% NEAA and 100 µg/mL gentamycinMEMM_G:BMEM_G additionally supplemented with 10 mM MgCl_2_

### 2.2. Viruses and Inocula

HEV-positive samples were surplus material from our diagnostic laboratory stored at −80 °C. Successfully from serum isolated HEV strains 14-16753 (2.8 × 10^6^ HEV RNA copies/mL, genotype (gt) 3c), 14-22707 (1.6 × 10^6^ c/mL gt 3e) and 15-22016 (4.0 × 10^8^ c/mL, gt 3f-like) were derived from a 59 year old male, a 65 year old female and a 63 year old male subject, respectively. HEV from plasma (77-year old male) positive for strain 13-14672 (4.3 × 10^3^ c/mL, gt 3 subtype not assignable) and from faeces (72-year-old male) positive for strain 14-16078 (1.1 × 10^4^ c/mL, gt 3e) could not be isolated in cell culture. No further information about the subjects regarding serology, immunosuppression or phase of HEV infection is available. HEV gt 3c strain 47832c positive culture supernatant served as positive control. The positive control was always freshly prepared together with subject specimens. Samples were diluted with PBS without Ca^2+^ and Mg^2+^ containing 0.2% BSA (Sigma-Aldrich) (*w*/*v*) (PBS^(−)^/BSA^0.2%^) which always served as a negative control. The faecal specimen was processed as a 10% faecal suspension in PBS^(−)^/BSA^0.2%^. Diluted specimens were thoroughly vortexed, centrifuged at 8000× *g* for 10 min and the supernatant sterile-filtrated using a 0.2 µm polyethersulfone (PES) membrane (Sarstedt, Nümbrecht, Germany). No further pretreatment was applied.

### 2.3. Virus Isolation and Passaging

Unless stated differently in the results section, isolation was carried out in T12.5 flasks as follows: BMEM cultured cells were seeded at a concentration of 10^5^ viable cells/cm^2^ in T12.5 flasks 14 days prior to inoculation and cultured at 37 °C and 5% CO_2_. Medium was switched from BMEM to MEMM and completely refreshed every 3–4 days. After 14 days, supernatant was replaced with 250 µL of inoculum and cells were incubated for 75 min at room temperature. Afterwards, 2.5 mL of MEMM were added and cells were incubated at 34.5 °C and 5% CO_2_. At 24 h later, supernatant was completely refreshed with MEMM and from then on every 3–4 days.

### 2.4. HEV RNA Quantification

RNA was isolated on an EZ1^®^ Advanced XL workstation using the EZ1 Virus Mini Kit v2.0 (Qiagen, Hilden, Germany). Eluted nucleic acid was tested by RT-qPCR according to Wenzel et al. [31]. HEV RNA was quantified as genome copies per mL (c/mL).

### 2.5. Detection of ORF2 Antigen

The commercially available HEV Ag ELISA Plus kit (Wantai, Beijing, China) was used for detecting ORF2 antigen. The ELISA was performed according to the manufacturer’s protocol.

### 2.6. Immunofluorescence Microscopy

Inoculated cell layers were detached after different time points, seeded in a 96-well µ-plate (Ibidi, Martinsried, Germany) and grown to ~80% confluency at 37 °C and 5% CO_2_ within two days. Afterwards, cells were washed with 0.05% Tween-20 (Sigma-Aldrich) (*v*/*v*) in PBS with Ca^2+^ and Mg^2+^ (PBS^(+)^, Lonza, Basel, Switzerland) (PBS^(+)^/Tween-20^0.05%^) and fixed with 2% formaldehyde (Merck, Darmstadt, Germany) (*v*/*v*) in PBS^(+)^. After another washing step, cells were permeabilized with 0.1% Triton X-100 (Sigma-Aldrich) (*v*/*v*) in PBS^(+)^ followed by blocking with 2% BSA (*w*/*v*) in PBS^(+)^/Tween-20^0.05%^. The blocking solution was then replaced by an anti-hepatitis E ORF2 antibody (clone 2E2) [32] (Merck) diluted in PBS^(+)^ containing 1% BSA (*w*/*v*) (PBS^(+)^/BSA^1%^) to a concentration of 5 µg/mL and incubated for one hour at room temperature. After washing, cells were incubated with an anti-mouse IgG_1_-FITC antibody (sc-2078, Santa Cruz Biotechnology, Dallas, TX, USA) diluted in PBS^(+)^/BSA^1%^ to a concentration of 2 µg/mL and incubated in the dark for one hour at room temperature. After washing, nuclei were stained by SlowFade^®^ Diamond Antifade Mountant with DAPI (Thermo Fisher Scientific, Waltham, MA, USA). Fluorescent images were taken with a Keyence BZ-9000 microscope.

### 2.7. Whole Genome Sequencing

Isolated RNA of strain 14-16753, 14-22707 and 15-22016 was reverse transcribed by MuLV (Applied Biosystems, Waltham, MA, USA) and amplified by a first round PCR in different overlapping parts (Appendix A). The cDNA synthesis of the 5′-end was carried out with the SuperScript™ III First-Strand Synthesis System kit (Thermo Fisher Scientific). Primers used for cDNA synthesis of the 5′- and 3′-end followed by a first round PCR are based on the BD SMART™ RACE cDNA Amplification kit protocol (Appendix A). All other primers are either based on Johne et al. [23] or were designed using Primer3 (http://primer3.ut.ee/) [33] and GEMI v1.5.1 [34]. PCR products were further amplified by a nested PCR. These products were separated on agarose gels and all amplification products were extracted using the QIAquick Gel Extraction kit (Qiagen). Purified amplicons were then sequenced on an ABI 3130xl automated sequencer. Electropherograms were inspected and sequences assembled with CodonCode Aligner v4.2.7 (www.codoncode.com, CodonCode Corporation, Centerville, MA, USA). All PCR reactions were performed with a Veriti™ 60-well Thermal Cycler (Thermo Fisher Scientific). Consensus sequences were deposited in the GenBank under accession numbers MK089849 (14-16753, gt 3c), MK089848 (14-22707, gt 3e) and MK089847 (15-22016, gt 3f-like).

### 2.8. Phylogenetic Analysis

Obtained whole genome sequences were genotyped by using the current HEV subtype reference set proposed by Smith et al. [3]. Multiple sequence alignments (msa) were carried out by using MUSCLE v3.8.31 [35]. Msa-files were further processed with RAxML v8.2.10 [36] and the best matching phylogenetic tree was calculated based on the maximum likelihood principle with a bootstrap of 1000 replicates. The tree was rooted using moose HEV [37] as an outgroup and visualized by FigTree v1.4.3 (http://tree.bio.ed.ac.uk/software/figtree/).

## 3. Results

### 3.1. Effect of Medium Refreshment on Cells Infected With HEV Strain 47832c and Isolation of HEV Gt 3c Strain 14-16753

Supernatant of A549 persistently infected with genotype (gt) 3c strain 47832c was examined after different time points. A maximum of HEV RNA accumulated after 64 h whereas HEV ORF2 antigen (Ag) concentrations kept on rising even if medium was not refreshed until 168 h (Figure 1a). These data indicate that there is no need for daily medium refreshment to reach high viral loads.

To analyse the effect of medium exchange frequency, supernatants were either half or completely refreshed in different intervals. After five weeks, higher HEV RNA concentrations were generated when medium was refreshed completely and more frequently (Figure 1b). However, refreshing medium once or twice a week still resulted in high HEV RNA concentrations.

The impact of medium refreshment on de novo infection of cell lines was also investigated by inoculating A549 and PLC/PRF/5 with isolate 47832c (A549 supernatant containing 2.1 × 10^7^ c/mL). In addition, HEV-positive material of three different subjects were inoculated: a faecal suspension 14-16078 (1.1 × 10^4^ c/mL, gt 3e), serum 14-16753 (2.8 × 10^6^ c/mL, gt 3c) and plasma 13-14672 (4.3 × 10^3^ c/mL, gt 3 subtype not assignable). For strain 47832c, no difference was detected between medium refreshment once, twice or five times a week. An exception was weekly refreshment of A549 resulting in lower HEV ORF2 Ag concentrations (Figure 1c,d). Cells inoculated with human specimens showed low antigen concentrations. However some elevated values were detected for the serum sample and A549 inoculated with the faecal suspensions and refreshed five times a week. After 7 weeks HEV RNA was only detectable in the supernatant of A549 and PLC/PRF/5 inoculated with serum (long-term cultivation shown in Figure 6a,b). This experiment represented the first successful isolation of the novel HEV gt 3c strain labelled 14-16753.

Taken together, complete medium refreshment once a week is appropriate for HEV-positive PLC/PRF/5 cells but not for A549. Therefore, complete medium refreshment twice a week was applied in the following experiments.

### 3.2. Overconfluently Grown Cells Seeded >1 Week Prior to Inoculation are More Permissive for Infection

Since an HEV strain derived from an experimentally infected pig exclusively replicated in three-dimensional cultures of PLC/PRF/5 [19], the effect of cell confluence on susceptibility to HEV was investigated. A549 and PLC/PRF/5 were seeded at different concentrations and time points prior to inoculation with strain 47832c (A549 supernatant containing 2.6 × 10^8^ c/mL). At 0 days prior to cell seeding, 2.5 mL of defined cell suspensions were transferred to T12.5 flasks and 250 µl of inoculum was immediately added before cells could adhere. Supernatants were tested for HEV RNA after 14 and 28 days post inoculation (dpi). Detection of HEV RNA at 28 dpi represented a successful infection. All A549 cultures were successfully infected with HEV (Figure 2a). Cells seeded at least 7-10 days (depending on cell concentration) prior to virus inoculation generated higher viral loads in shorter periods. This trend was clearly more pronounced in PLC/PRF/5 (Figure 2b), which could only be reproducibly infected when seeding cells at the default split ratio of 1:4 at least 14 days before inoculation. This lead time can be shortened to 10 and 7 days by seeding 1 × 10^6^ to 9 × 10^6^ viable cells, respectively. The fact, that viral load at 28 dpi does not exceed the inoculum’s load (except for PLC/PRF/5 seeded 35 days prior to inoculation) may be due to the slow replicating nature of HEV.

Generally, we observed a tendency to more reliable infection and higher viral loads when more cells are seeded earlier prior to inoculation, generating more susceptible three-dimensional cell layers. Therefore and to standardize the approach, cell seeding was set to 1 × 10^5^ viable cells per cm^2^ at 14 days prior to inoculation.

### 3.3. Distinct Medium Supplements Enhance HEV Replication and Optimized Media Promote De Novo Isolation of HEV Gt 3e Strain 14-22707

The concentration of supplemented FCS is usually reduced in medium used for virus culturing. Therefore, A549 persistently infected with isolate 47832c were maintained in BMEM_G containing different concentrations of heat-inactivated FCS. The supernatant was completely refreshed and tested for HEV ORF2 Ag after different periods. Incubating HEV-positive cells with higher concentrations of FCS resulted in higher HEV ORF2 Ag production (Figure 3a).

Previous HEV isolation trials used medium supplements amphotericin B and 30 mM MgCl2 [14] or non-essential amino acids (NEAA) [23] and different antibiotics. In addition, a serial dilution experiment revealed supplementation of 10 mM CaCl_2_, KCl, K_2_SO_4_, MgCl_2_, MgSO_4_ or Na_2_SO_4_ to enrich HEV ORF2 Ag concentrations in the supernatant whereas KH_2_PO_4_, NaCl and Na_2_HPO_4_ decreased antigen concentrations (data not shown). Therefore, A549 and PLC/PRF/5 maintained in differently supplemented BMEM_G were inoculated with isolate 47832c-containing A549 supernatant (2.6 × 10^8^ c/mL). Subsequent supernatants were tested for HEV RNA after 14 dpi and ORF2 Ag after 28 dpi. Medium supplements enhancing HEV replication most distinctly are shown in Figure 3b. Surprisingly, supplementing 2.5 µg/mL amphotericin B resulted in the highest increase of HEV RNA and ORF2 Ag, at least in PLC/PRF/5 supernatant. Supplementation of salts revealed K_2_SO_4_ combined with CaCl_2_ to enhance HEV replication most pronouncedly. However, cell viability decreased by ~50% in presence of supplemented CaCl_2_. Viral loads were also increased in PLC/PRF/5 but not in A549 by supplementing NEAA. Adding 100 U/mL penicillin G and 100 µg/mL streptomycin resulted in slightly higher HEV RNA and ORF2 Ag levels compared to 100 µg/mL gentamycin (data not shown).

Based on these results, A549, PLC/PRF/5, HepG2/C3A and HuH-7-Lunet BLR either maintained with BMEM, MECK (BMEM supplemented with 2.5 µg/mL amphotericin B, 10 mM CaCl_2_ and 10 mM K_2_SO_4_) or MEMM (adapted from Tanaka et al. [14]; BMEM supplemented with 2.5 µg/mL amphotericin B and 30 mM MgCl_2_) were inoculated with PLC/PRF/5 supernatant containing isolate 47832c (5.7 × 10^6^ c/mL), isolate 14-16753 (8.3 × 10^6^ c/mL) or with an HEV gt 3e-positive serum (1.6 × 10^6^ c/mL). At 49 dpi, higher viral loads were always found in the supernatant of cells cultivated with MECK or MEMM compared to BMEM (Figure 3c). HEV gt 3e was successfully isolated from serum in PLC/PRF/5 and A549 irrespective of the medium. In contrast, HEV gt 3e could only be isolated in HuH-7-Lunet BLR maintained in optimized medium and HepG2/C3A was not susceptible to HEV gt 3e at all. The novel isolate was labelled 14-22707 (long-term cultivation shown in Figure 6c). MEMM showed better results than MECK in three out of four cell lines and was therefore selected as the culture medium of choice.

### 3.4. First Isolation of HEV Gt 3f-like Strain 15-22016 Confirms PLC/PRF/5 as the Most Suitable Cell Line for Culturing HEV

Many different cell lines have already been tested for their capability of supporting the replication of HEV in vitro. A549, its subclone A549/D3, PLC/PRF/5 and HepG2/C3A were found to be the most susceptible cell lines to HEV [14,21,22,30]. However, these cell lines were never systematically compared with different HEV strains. In addition, the cell clone HuH-7-Lunet BLR might also be a potential candidate for HEV since it is highly permissive to hepatitis C virus replication [29].

Therefore, these cells were inoculated with PLC/PRF/5 supernatant containing isolate 47832c (8.0 × 10^7^ c/mL) or 14-16753 (4.2 × 10^8^ c/mL). Figure 4 shows that isolate 47832c replicated fastest in A549/D3 and reached maximum viral loads at 14 dpi (tenfold higher as in A549). However, at 29 dpi PLC/PRF/5 produced comparable amounts of HEV. In contrast, isolate 14-16753 always replicated faster and to higher viral loads in PLC/PRF/5.

This trend was confirmed by inoculating PLC/PRF/5 supernatant containing isolate 47832c (1.3 × 10^8^ c/mL), 14-16753 (6.0 × 10^8^ c/mL), 14-22707 (4.0 × 10^8^ c/mL) or serum containing gt 3f-like HEV (1.1 × 10^8^ c/mL) onto A549, PLC/PRF/5, HepG2/C3A and HuH-7-Lunet BLR. Again, PLC/PRF/5 generated the highest viral loads in >80% of all time points and strains. Compared to other cell lines the median HEV RNA concentration in PLC/PRF/5 supernatant was approximately 10-, 100- and 4000-fold increased at 14, 28 and 49 dpi, respectively. Two exceptions were observed with isolate 14-16753 which replicated better in Huh-7-Lunet BLR at 14 and 49 dpi. The novel HEV gt 3f-like strain 15-22016 was successfully isolated in PLC/PRF/5, Huh-7-Lunet BLR and HepG2/C3A but not in A549 (long-term cultivation shown in Figure 6d).

Therefore, PLC/PRF/5 as a three-dimensional cell layer is the most suitable host for isolating and maintaining HEV in vitro. The newly introduced cell clone HuH-7-Lunet BLR also clearly supports the replication of HEV and generates higher viral loads than A549 and HepG2/C3A.

### 3.5. Titration and Growth of Newly Isolated HEV Strains Reveals Different Replication Kinetics

PLC/PRF/5 were inoculated with serial tenfold dilutions of first-passage isolate 14-16753 (gt 3c), 14-22707 (gt 3e) and 15-22016 (gt 3f-like). Isolate 14-22707 replicated fastest and increased its viral load tenfold within 4.8 days (Figure 5b). Isolate 14-16753 increased its viral load tenfold in 6.3 days and isolate 15-22016 in 8.5 days (Figure 5a,c). These values are based on a linear regression with the logarithm of HEV RNA concentrations from 4–21 dpi. Isolate 14-22707-positive cells showed a brighter fluorescence compared to isolates 14-16753 and 15-22016 after immunofluorescence staining. Isolate 15-22016 resulted in relatively weak signals. This could be due to the application of a monoclonal anti-ORF2 antibody (clone 2E2 [32]).

The 50% tissue culture infective dose (TCID_50_) was determined in a subsequent titration experiment. PLC/PRF/5 were inoculated in triplicates in T12.5 flasks with tenfold serial diluted viral stocks. Supernatants were tested for HEV RNA at 13 dpi. T12.5 flasks with detectable HEV RNA were defined as “infected” and T12.5 with no detectable HEV RNA as “uninfected”. TCID_50_ was calculated using the Reed and Muench method resulting in 1.3 × 10^3^ TCID_50_/mL for isolate 47832c (viral load of stock was 2.4 × 10^8^ c/mL) and 14-16753 (1.6 × 10^8^ c/mL). For isolate 14-22707 (8.7 × 10^8^ c/mL) and 15-22016 (1.1 × 10^8^ c/mL) the TCID_50_ was 2.2 × 10^3^ TCID_50_/mL.

### 3.6. Long-Term Cultivation of Persistently Infected Cells Leads to Virtually Unlimited Production of HEV

The novel isolated HEV strains together with isolate 47832c were followed up for a period of 30 weeks to more than 2 years, whereby no cytopathic effect was ever observed. The highest viral loads detected were 7.4 × 10^8^ c/mL (208 dpi) and 1.5 × 10^9^ c/mL (476 dpi) for isolation and first passage of strain 14-16753, respectively (Figure 6a,b). The maximum of isolate 14-22707 and 15-22016 was detected at 134 dpi with 4.4 × 10^9^ c/mL and at 274 dpi with 5.5 × 10^9^ c/mL, respectively (Figure 6c,d). Remarkably, cells could be maintained for more than a year in one single T12.5 flask by solely refreshing the medium completely every 3-4 days. Over the whole period, infectious HEV was secreted into the medium resulting in high viral loads. All isolates were successfully passaged. Moreover, HEV positive cells were successfully expanded, frozen, thawed and re-cultured again and continued to produce infectious HEV. Surprisingly, the viral load of the first passage of isolate 14-16753 constantly decreased when cells were split regularly (Figure 6b). In contrast, this was not observed with isolate 14-22707 (Figure 6c) and will be investigated further in future experiments.

### 3.7. Characterization of Whole Genome Sequences of Novel Isolates Revealed No Insertion in ORF1

Viral RNA isolated from supernatant (strain 14-16753 and 14-22707: PLC/PRF/5 at 200 dpi and 189 dpi, respectively; strain 15-22016: HuH-7-Lunet BLR at 126 dpi) was reverse transcribed to cDNA followed by amplification of overlapping PCR fragments and Sanger sequencing. Sequences were compared to all hitherto cell culture-isolated and whole genome-sequenced strains (47832_serum, 47832c_p2 [23], LBPR-0379_serum [22], Kernow-C1_faeces, Kernow-C1_p6 [21], HE-JF5_serum [38], HE-JF5/15F_p6 [15], JE03-1760F_faeces [39] and JE03-1760F_p0 [14]). Newly isolated strains clearly clustered with subtypes 3c, 3e and 3f reference strains (Figure 7). Unlike the recently isolated strains 47832c, LBPR-0379 and Kernow-C1, the new isolates do not harbour insertions in the HVR of ORF1 which was shown to be associated with a growth advantage in vitro [40]. This is the first time that HEV isolates of the 3efg-clade could be propagated in cell culture and replicated to high viral loads of >10^9^ c/mL.

### 3.8. Optimized Method for Successful HEV Isolation from Serum

According to the results of the study, an optimized protocol for HEV isolation from serum in cell culture should include: Seeding PLC/PRF/5 in T12.5 flasks at a concentration of 10^5^ viable cells per cm^2^. Maintaining the cells in MEMM at 37 °C and 5% CO_2_ for two weeks and refreshment of medium every 3–4 days. If necessary, diluting HEV positive serum to an adequate volume with PBS without Ca^2+^ and Mg^2+^ containing 0.2% BSA (*w*/*v*). Vortexing, centrifuging at 8000× *g* for 10 min and sterile-filtering using a 0.2 µm PES membrane. No further pretreatment is needed. Afterwards, the inoculum should at least contain 10^5^ HEV RNA copies/mL. The three-dimensional cell layer was then inoculated with 250 µL per T12.5 for 75 min at room temperature. 2.5 mL MEMM was then added and incubated at 34.5 °C and 5% CO_2_. The medium was completely refreshed 24 h later and every 3–4 days afterwards.

## 4. Discussion

Several approaches to HEV cell culture have already been described. However, these approaches could either not fully be reproduced [18,19,20], could only isolate strains carrying an uncommon insertion in ORF1 [21,22,23] or need a sophisticated setup to generate three-dimensional cultures [19]. This work aimed to improve the published approaches to HEV cell culture, make it an easy-care system which can be continuously maintained for virtually unlimited periods of time and isolate novel wild-type HEV strains.

Optimization experiments led to the isolation of three novel strains labelled 14-16753 (genotype (gt) 3c), 14-22707 (gt 3e) and 15-22016 (gt 3f-like). These isolates replicate to high viral loads of >10^9^ c/mL and differ in replication kinetics. Moreover, this is the first time that whole genome sequences of two stably cell culture propagated isolates of clade 3efg were determined. In addition, the three new isolates represent the predominant subtypes 3c, 3e and 3f currently circulating in Europe.

Our results show that (i) maintenance effort can be decreased substantially by reducing medium refreshment from daily to twice a week, (ii) overconfluent (three-dimensional) cell layers grown in flasks are more susceptible to HEV compared to usual monolayers, (iii) medium supplements, especially amphotericin B, MgCl_2_, CaCl_2_ and K_2_SO_4_ increase HEV replication in vitro, (iv) PLC/PRF/5 as a three-dimensional cell layer is the most permissive cell line to HEV and HuH-7-Lunet BLR also substantially supports HEV replication, (v) HEV-producing cells can be kept in culture for >1 year with only medium refreshment every 3–4 days and still generate very high viral loads of ~10^9^ c/mL.

Surprisingly, the susceptibility of overconfluently grown cells to HEV infection depends on the cell line. For instance, A549 can be seeded at virtually any time at almost any concentration and still support HEV replication. However, higher viral loads are achieved faster with mature A549 cell layers. This effect is more pronounced in PLC/PRF/5 which must be seeded ≥14 days prior to inoculation at their default split ratio to guarantee HEV infection. The lead time of 14 days can be reduced to as low as 0 if the concentration of seeded cells is increased. For a reliable infection, however, 7 days of cell growing are still needed to generate a susceptible overconfluent cell layer forming three-dimensional structures. This stands in line with Berto et al. who observed that solely differentiated PLC/PRF/5 grown as three-dimensional cultures support HEV replication but not common monolayers [19]. Cell differentiation and closer contacts between cells in the three-dimensional cell layers may result in a higher susceptibility to HEV. In addition, autophagy processes may be considered since overconfluent cell layers consisting of much more cells compared to monolayers, were maintained in the same amount of medium and may therefore run low on nutrients.

Medium supplements are also important for HEV isolation and maintenance. In 1999, Huang et al. reported that 30 mM MgCl_2_ supplemented to the maintenance medium increased HEV loads and preserved infectivity [41]. We found that not only MgCl_2_ but also 10 mM CaCl_2_, KCl, K_2_SO_4_, MgSO_4_ and Na_2_SO_4_ benefit to HEV replication whereas KH_2_PO_4_, NaCl and Na_2_HPO_4_ adversely affect the replication. The FCS concentration also influences the potential of HEV replication in vitro. Commonly, 2% FCS are added to a virus maintaining medium. However, for HEV, we found that 10% FCS was more appropriate. The medium supplement with the highest impact on viral loads was found to be amphotericin B. Noteworthy, amphotericin B also promotes influenza virus replication in cell culture [42]. The authors hypothesized that the antimycotic drug contributes to the acidification of internal cell compartments which in turn promote fast pH decrease within endosomes and therefore virus infectivity. Since HEV enters liver cells via clathrin-mediated endocytosis [43] and HEV infectivity depends on acidification of endosomes [44] these facts would be in line with the observed influence of amphotericin B on HEV replication in vitro.

HEV was described to replicate in several cell lines and A549, PLC/PRF/5 and HepG2/C3A were the most promising ones [14,21]. Therefore, these cell lines were investigated for HEV replication competence together with cell clones A549/D3 (supports HEV strain 47832c replication more efficiently [30]) and HuH-7-Lunet BLR (highly permissive for hepatitis C virus replication [29]). Until 14 dpi it is strain-dependent which cell line generates the highest viral loads. But as of week four post inoculation PLC/PRF/5 always generate the highest viral loads independent of isolation or passaging of a strain. High viral loads were also generated with HuH-7-Lunet BLR which were generally higher compared to A549 and HepG2/C3A.

The latest propagated HEV strains 47832 [23], LBPR-0379 [22] and Kernow-C1 [21] were isolated from chronically infected patients and harbour an insertion in the ORF1 either acquired by recombination [21,22] or derived from its own ORF1 [23]. HEV genomes of Kernow-C1 harbouring an insertion in ORF1 were classified as a minor species [21] and became the predominant species in cell culture [22]. Moreover, the insertion confers a growth advantage in vitro [40]. Our isolates do not harbour insertions in ORF1. This indicates that our optimized cell culture system is not only permissive to adapted minor species but also to wild-type dominant species.

Our isolates have been kept in culture continuously for 1 to >2.5 years and persistently infected cells still produced viral loads of ~10^9^ c/mL. Presumably, HEV-positive cells can be continuously maintained for a virtually unlimited period of time. This is an important aspect since immunocompromised patients chronically infected with HEV rely on an effective antiviral treatment. However, only off-label drugs such as ribavirin and sofosbuvir are currently available. While treatment with ribavirin leads to a sustained virologic response in three-quarters of patients [45] there are already numerous ribavirin-associated mutations described [46] which lead to treatment failure [47]. Sofosbuvir was shown to inhibit HEV Kernow-C1 replication in vitro and the combination with ribavirin even resulted in an additive effect [48]. However, there is not enough clinical data yet to estimate the effectivity in vivo due to few contradictory reports [49,50,51]. Recent reports suggest silvestrol as a drug candidate to treat HEV [52,53]. Our long-term HEV producing cell culture model may step in here and serve as a more lifelike setting to test drug candidates in vitro.

There are several uncertainties and possible limitations. First, HEV replicates very slowly to adequate viral concentrations after 1–10 weeks. This may hinder fast application of down-stream processes. Second, due to the absence of a cytopathic effect, replication has to be specifically confirmed by consecutive PCR testing or viral antigen detection. Third, it needs to be examined, if HEV of genotypes other than gt 3 can be propagated with comparable efficiency in the optimized cell culture system. Fourth, our cell culture system is based on a continuous cell line that may not reflect cells in vivo. Fifth, the question arises if and how the genome of the isolated strains changes after long-term cultivation or repeated passaging. Finally, our cell culture system does not include animal-derived cells which may be more useful for studying zoonotic aspects of HEV.

In summary, we isolated three novel HEV strains of the predominant subtypes in Europe stably replicating to high viral loads while optimizing and simplifying the cell culture system. This system may be useful for studies on the HEV life cycle, inactivation, drug and vaccine development.

## Figures and Tables

**Figure 1 viruses-11-00483-f001:**
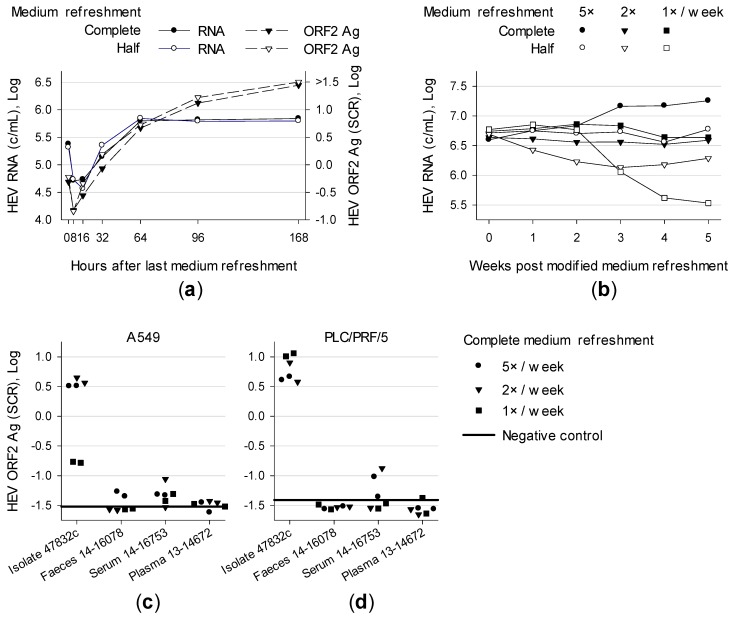
Effect of medium refreshment of cells persistently or de novo infected with HEV. (**a**) Supernatant of A549 persistently infected with HEV strain 47832c was refreshed either half or completely after lengthening time points and tested for HEV RNA and ORF2 antigen (Ag). (**b**) Supernatant of A549 persistently infected with HEV strain 47832c was either changed half or completely once, twice or five times a week and tested weekly for HEV RNA. (**c**) A549 and (**d**) PLC/PRF/5 were inoculated in technical duplicates with isolate 47832c as well as with HEV-positive materials from three different patients, namely a faecal suspension 14-16078, serum 14-16753 and plasma 13-14672. After inoculation, medium was changed completely once, twice or five times a week and supernatants were tested for HEV ORF2 Ag 24 days post-inoculation (dpi). SCR, signal-to-cut-off ratio.

**Figure 2 viruses-11-00483-f002:**
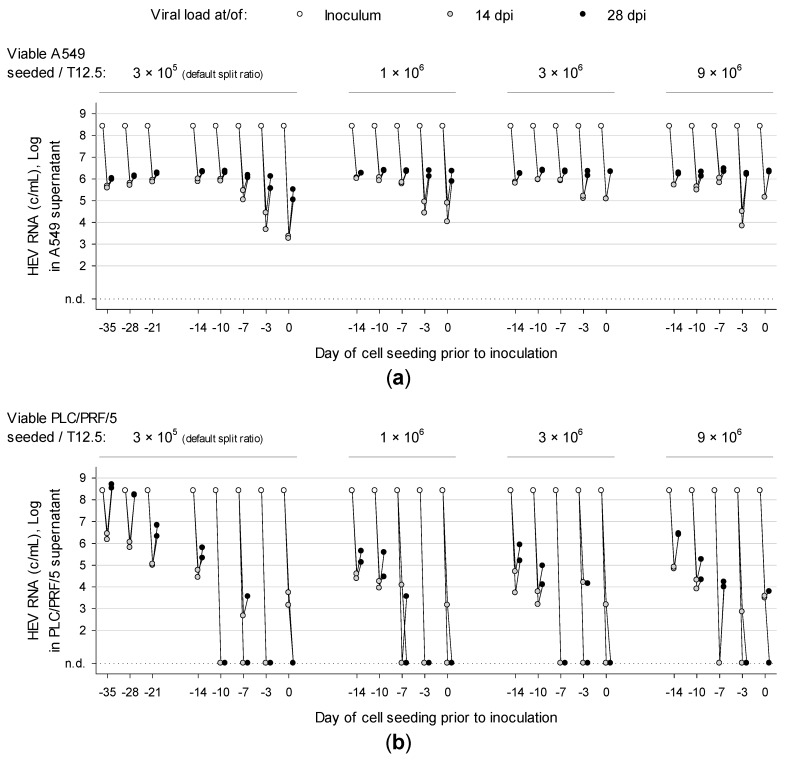
Density and time point of cell seeding prior to inoculation affect the success of infection and consequent viral load. (**a**) A549 and (**b**) PLC/PRF/5 were seeded at different concentrations and time points prior to inoculation. Cells were inoculated in duplicates with isolate 47832c. Supernatants were tested for HEV RNA at day 14 and 28 post-inoculation. Dpi, days post inoculation; n.d., not detected.

**Figure 3 viruses-11-00483-f003:**
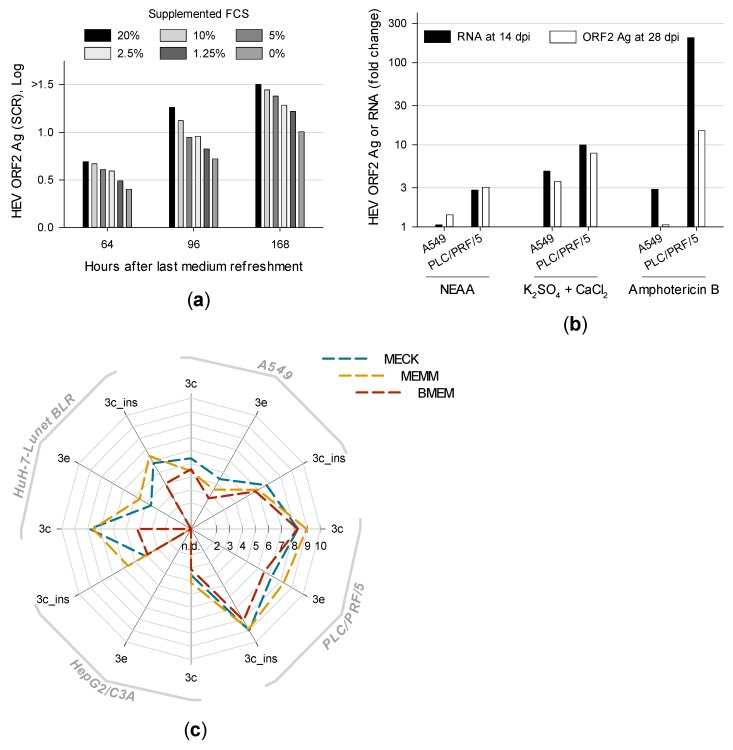
Distinct medium supplements and the combinations thereof increase viral loads and optimized media promote infection. (**a**) A549 persistently infected with HEV strain 47832c were maintained with medium supplemented with different concentrations of heat-inactivated fetal calf serum (FCS). After 64, 96 and 168 h supernatants were refreshed and tested for ORF2 antigen (Ag). (**b**) A549 and PLC/PRF/5 maintained with differently supplemented media were inoculated with isolate 47832c positive A549 supernatant and subsequent supernatants were tested for HEV RNA at 14 days post inoculation (dpi) and for ORF2 Ag at 28 dpi. Results are shown as the mean fold change of technical duplicates. (**c**) A549, PLC/PRF/5, HepG2/C3A and HuH-7-Lunet BLR were maintained in optimized media MECK (10 mM CaCl_2_, 10 mM K_2_SO_4_, amphotericin B) and MEMM (30 mM MgCl_2_ and amphotericin B) as well as in basic culture medium BMEM. Cells were inoculated with PLC/PRF/5 supernatant containing either isolate 47832c (marked as 3c_ins) or 14-16753 (marked as 3c) or with serum 14-22707 positive for HEV gt 3e (marked as 3e). After 49 dpi supernatants were tested for HEV RNA. Results are shown as the mean of technical triplicates. N.d., not detected; NEAA, non-essential amino acids.

**Figure 4 viruses-11-00483-f004:**
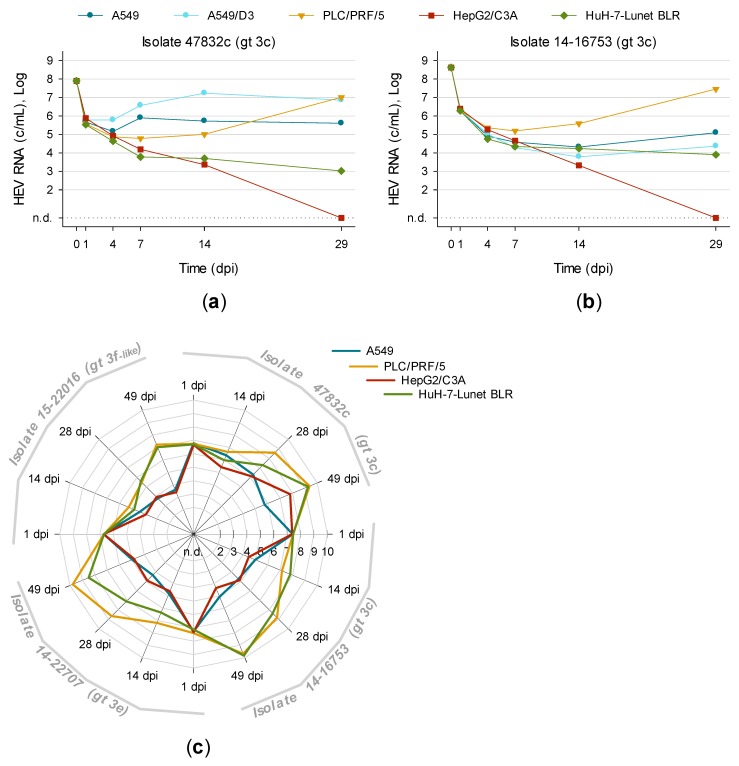
Effect of cell lines on de novo infection with different HEV strains. A549, its subclone A549/D3, PLC/PRF/5, HepG2/C3A and HuH-7-Lunet BLR were inoculated with either (**a**) isolate 47832c or (**b**) isolate 14-16753. Supernatants were tested for HEV RNA at 1, 4, 7, 14 and 29 days post inoculation (dpi). Results are shown as the mean of technical triplicates. (**c**) A549, PLC/PRF/5, HepG2/C3A and HuH-7-Lunet BLR were inoculated with isolate 47832c, 14-16753 or 14-22707 as well as with serum 15-22016 positive for HEV gt 3f-like. After 1, 14, 28 and 49 dpi supernatants were tested for HEV RNA. N.d., not detected.

**Figure 5 viruses-11-00483-f005:**
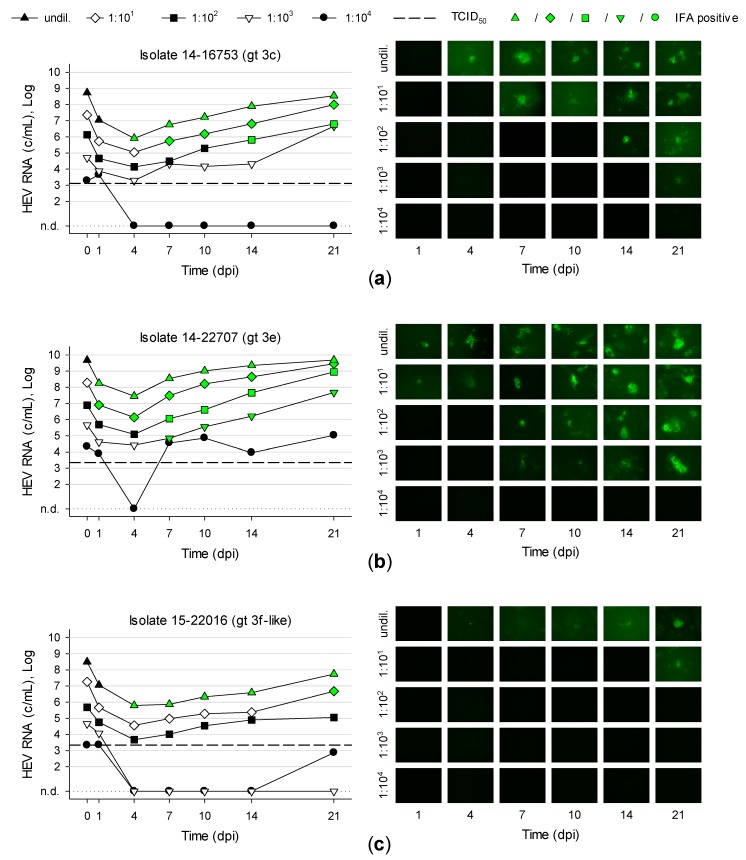
Titration and replication kinetics of novel isolates. PLC/PRF/5 were inoculated with serial tenfold dilutions of first passage isolates (**a**) 14-16753, (**b**) 14-22707 and (**c**) 15-22016. After 1, 4, 7, 10 14 and 21 days post inoculation (dpi) supernatants were tested for HEV RNA (left panels) and cells were indirectly stained for HEV capsid antigen by immunofluorescence assay (IFA; right panels). Dashed lines represent the respective TCID_50_ values determined in a separate experiment. N.d., not detected, undil., undiluted.

**Figure 6 viruses-11-00483-f006:**
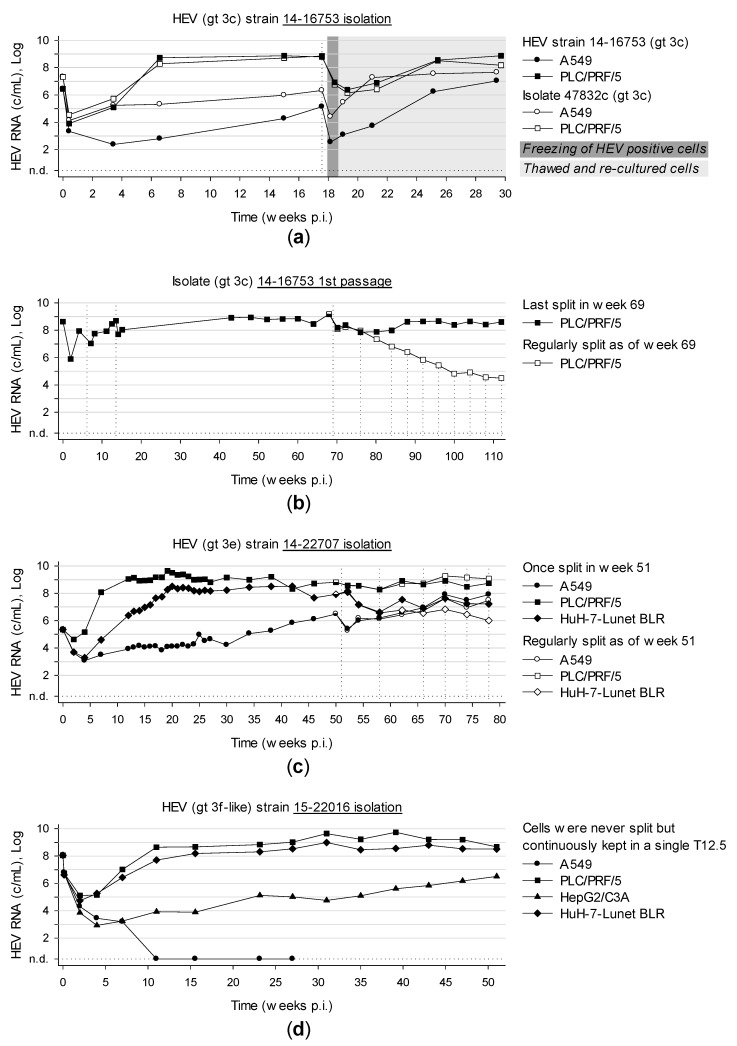
Long-term cultivation of cells persistently infected with HEV. (**a**) A549 and PLC/PRF/5 cells were inoculated with isolate 47832c and strain 14-16753-positive serum. After 123 dpi, cells were expanded, frozen and transferred to liquid nitrogen. Frozen cells were re-cultured and further followed up. (**b**) PLC/PRF/5 cells were inoculated with isolate 14-16753-positive PLC/PRF/5 supernatant. After 51 weeks, the effect of regular splitting on HEV load was investigated. (**c**) A549, PLC/PRF/5 and HuH-7-Lunet BLR cells were inoculated with strain 14-22707-positive serum and followed up for 1.5 years. (**d**) A549, PLC/PRF/5, HepC2/C3A and HuH-7-Lunet BLR cells were inoculated with serum positive for strain 15-22016 and followed up for one year. Culture supernatants were tested for HEV RNA. Vertical dotted lines mark passaging of cells. N.d., not detected; p.i., post inoculation.

**Figure 7 viruses-11-00483-f007:**
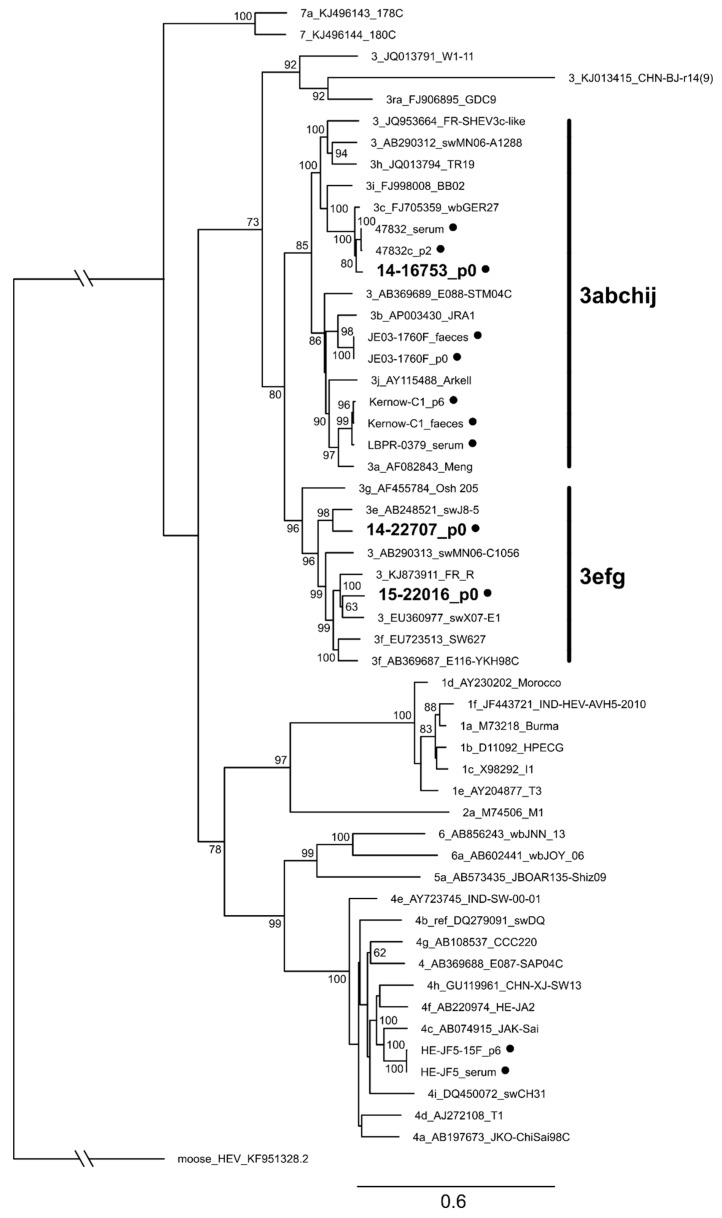
Phylogenetic relationship of HEV strains isolated in cell culture together with reference strains. The tree is based on complete genome sequences and was rooted by using moose HEV as an outgroup. Sequences of the HEV genotype reference strains as proposed by Smith et al. [3] were aligned together with the new isolates as well as with all HEV strains isolated in cell culture with available whole genome sequences as of April 2019. Branches with a bootstrap value of >60% were labelled. Reference sequences are denoted by subtype, GenBank accession number and strain name. Strains isolated in cell culture are marked with black circles and denoted with strain name and source material or number of passages (p). Novel isolates are shown in bold. Bold vertical lines denote subclades 3abchij and 3efg of genotype 3.

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
