# Peer review of "Isolation of Subtype 3c, 3e and 3f-Like Hepatitis E Virus Strains Stably Replicating to High Viral Loads in an Optimized Cell Culture System"

_viruses, 2019, doi:10.3390/v11060483_

Round 1
Reviewer 1 Report
The paper written by Schemmerer et al. interested in the optimization of cell culture systems for HEV. This is a very interesting topic since to date, culturing HEV in vitro is not easy. The authors tested several cell lines and culture media. Overall, the data is interesting and will be helpful for the community. However, I have some concerns to address.
Major points :
1. « the products were purified on an agarose gel and bands of expected sizes were extracted… » (lines 139-140) […] « the new isolates do not harbour insertions in the HVR of ORF1 » (line 347) and « our isolates do not harbour insertions in ORF1 » (lines 420-1). According to the protocol the authors used, how to be sure they do not miss some recombinant strains since only the « bands of expected sizes » were sequenced ? This is important to claim that the isolates do not harbour insertions.
2. Why do the authors follow-up the Ag concentration during infection ? In humans infected with HEV, it was described that the Ag can be detected despite the absence of HEV RNA. Thus, it is possible that the detection of HE Ag does not preclude the good efficiency of HEV RNA replication.
3. The authors presented 3 strains that grow in their system, which is very nice. However, it would be very interesting to know how many strains were screened to determine whether the authors only kept these strains among a lot of samples or if they just tested these strains. In addition, when do the samples were collected in the patients ? At the acute phase ? Were the patients immunocompromised ? Were the samples tested for IgG and IgM ?
4. Since culture from serum or plasma samples is tricky, could the authors give more details about samples pretreatment (for instance centrifugation, any purification protocol ?). This would be very helpful for the readers who want to reproduce these experiments.
Minor points :
« belonging to the species Orthohepevirus A » (line 33) : this is not accurate enough. The family and the existence of several species should be mentionned.
« ORF1 encodes for non structural proteins » (line 39) : to date, it is unknown whether the protein is cleaved or not. Maybe the authors could replace by « encodes the non structural polyprotein ».
« ORF2 for the capsid protein » (line 39) : another reference could have been chosen. In addition, for the readers non familiar with HEV, the introduction should mention the existence of the free Ag also encoded by the ORF2. The free form is the main one detected by the assay used by the authors for the detection of Ag. References could be Montpellier et al. Gastroenterology 2018, and Yin et al. PNAS 2018.
« to generate infectious HEV [20] » (line 55) : several teams used such system, please cite a review if possible rather than a study from a particular group.
« Rely on transfection rather than real infection » (lines 57-58) : real infection can be performed after stock preparation based on transfection. Thus, this sentence should be softened since transfection just represents one possibility.
The authors should harmonize the name of the strain 3f-like (line 19 ;96…). It is sometime written gt3 (for instance line 169).
It looks as if one panel is missing in the figure1. Indeed, the figure 1c is not related to HEV RNA but Ag (line 164) and the figure 1e does not exist (line 172).
The name of the media must be added in the figure 3b to help the reader.
Please explain why the strain 3e was not tested on HepG2C3A (figure 3c).
Figure 5, last left panel (gt3f) : it is surprising the 10 -4 dilution grows while the 10-3 does not.
Author Response
Response to Reviewer 1 Comments
The paper written by Schemmerer et al. interested in the optimization of cell culture systems for HEV. This is a very interesting topic since to date, culturing HEV in vitro is not easy. The authors tested several cell lines and culture media. Overall, the data is interesting and will be helpful for the community. However, I have some concerns to address.
Response: We thank the reviewer for the positive view on our manuscript.
Major points:
Point 1: « the products were purified on an agarose gel and bands of expected sizes were extracted… » (lines 139-140) […] « the new isolates do not harbour insertions in the HVR of ORF1 » (line 347) and « our isolates do not harbour insertions in ORF1 » (lines 420-1). According to the protocol the authors used, how to be sure they do not miss some recombinant strains since only the « bands of expected sizes » were sequenced ? This is important to claim that the isolates do not harbour insertions.
Response 1: We thank the reviewer for pointing us to this unintended inaccuracy in our text. We retrospectively re-checked all our archived gel images for the presence of multiple bands. We found no indication of insertions. Moreover, we took a detailed look on the PCR products and subsequent sequences covering the HVR:
- The HVR of isolate 14-16753 was completely covered by sequence segment 3n/12 and partly by 4n/12 (Table S1).
- The HVR of isolate 14-22707 was completely covered by sequence segment 3n/12 and partly by Hn11 and Gn11 (Table S2).
- The HVR of isolate 15-22016 was completely covered by sequence segments 1542n and 2001n (Table S3).
As above, no insertions (or deletions) were identified. To also make this fact clearer to the reader, we changed the misleading wording as follows: “These products were separated on agarose gels and all amplification products were extracted…” (lines 159-161). We hope this will clarify that we actually sequenced all resulting products and not only those of certain expected sizes.
Point 2: Why do the authors follow-up the Ag concentration during infection ? In humans infected with HEV, it was described that the Ag can be detected despite the absence of HEV RNA. Thus, it is possible that the detection of HE Ag does not preclude the good efficiency of HEV RNA replication.
Response 2: We tested for ORF2 Ag as a second readout besides PCR based RNA detection as a surrogate for active translation of viral nucleic acid to capsid proteins. It is true that ORF2 Ag in humans infected with HEV is peripherally longer detectable than HEV RNA. However, we think this observation cannot easily be transferred to our cell culture model which lacks an immune system and therefore exhibits virtually unlimited viral replication. Therefore, we think that an efficient HEV RNA replication results in robust expression of ORF2 Ag in cell culture.
Point 3: The authors presented 3 strains that grow in their system, which is very nice. However, it would be very interesting to know how many strains were screened to determine whether the authors only kept these strains among a lot of samples or if they just tested these strains. In addition, when do the samples were collected in the patients ? At the acute phase ? Were the patients immunocompromised ? Were the samples tested for IgG and IgM?
Response 3: We agree, this is certainly an important point for the reader, which we regrettably omitted in our first manuscript version. Besides the three isolated strains, we failed to isolate HEV from a faecal suspension 14-16078 (1.1 × 104 c/mL, gt 3e) and plasma 13-14672 (4.3 × 103 c/mL, gt 3 subtype not assignable). This was most probably due to the low viral load. To provide this information to the reader, we extended section “2.2. Viruses and Inocula” (lines 106-114).
The study design allowed including leftover samples from our diagnostic laboratory. This design and national legal regulation did not allow us to record personal or clinical patient information with the exception of gender and age, which is included in the respective section. The samples were most probably from acutely infected, non-immunocompromised patients, since the national reference lab for HEV is mainly analysing samples from this patient group. Moreover, subsequent testing for HEV antibodies unfortunately is not possible since the surplus material was completely used for inoculation of cell cultures.
Point 4: Since culture from serum or plasma samples is tricky, could the authors give more details about samples pretreatment (for instance centrifugation, any purification protocol ?). This would be very helpful for the readers who want to reproduce these experiments.
Response 4: We thank the reviewer for raising this important point about pretreatment, which was also issued by reviewer 2. We included this information (lines 119-121) to improve reproducibility and clarity of the manuscript. In this context, we also added the inoculum preparation procedure to the short summary of our cell culture procedure. The short summary was in the discussion section (original manuscript, lines 380-384) and we moved it –in line with suggestions by reviewer 3– to the results section and introduced a new subitem for that purpose (lines 390-400).
Minor points:
« belonging to the species Orthohepevirus A » (line 33) : this is not accurate enough. The family and the existence of several species should be mentionned.
Response: In the light of recent reports pointing out that rat HEV (Orthohepevirus C) may be an underestimated risk to humans we agree with the reviewer’s comment. We now included the other species in the text (lines 33-34) in order to provide the reader with a broader context.
« ORF1 encodes for non structural proteins » (line 39) : to date, it is unknown whether the protein is cleaved or not. Maybe the authors could replace by « encodes the non structural polyprotein ».
Response: We thank the reviewer for this suggestion and changed the text accordingly (lines 41-42).
« ORF2 for the capsid protein » (line 39) : another reference could have been chosen. In addition, for the readers non familiar with HEV, the introduction should mention the existence of the free Ag also encoded by the ORF2. The free form is the main one detected by the assay used by the authors for the detection of Ag. References could be Montpellier et al. Gastroenterology 2018, and Yin et al. PNAS 2018.
Response: We agree. The free form of ORF2 Ag is definitely a relevant point which should be mentioned. We added the two suggested references together with the sentence “Of note, ORF2 is translated into different forms of capsid protein and only a minority is associated with viral particles whereas the free form is abundantly secreted” (lines 42-44).
« to generate infectious HEV [20] » (line 55) : several teams used such system, please cite a review if possible rather than a study from a particular group.
Response: We agree and referenced the latest review on HEV cell culture by Meister et al. 2019 (line 63). In the context of reference [20] ([25] in revised manuscript) Gouttenoire et al. 2018, we wanted to point out that the infectious HEV clone G3-HEV83-2-27 is also a valuable tool in the research on HEV. However, we did not clearly name the strain and therefore added a separate sentence after listing the transfection approach to keep the Gouttenoire reference (lines 63-64).
« Rely on transfection rather than real infection » (lines 57-58) : real infection can be performed after stock preparation based on transfection. Thus, this sentence should be softened since transfection just represents one possibility.
Response: This point is well taken. To prevent (unintended) misunderstandings, we deleted the “rather than real infection” part from the text (lines 66-67).
The authors should harmonize the name of the strain 3f-like (line 19 ;96…). It is sometime written gt3 (for instance line 169).
Response: Line 169 (revised manuscript: lines 189-190) lists specimens positive for HEV other than the specimen positive for gt 3f-like HEV 15-22016. To avoid confusion, we added the specimen numbers after the materials and the fact, that the subtype of HEV in plasma 13-14672 was not assignable. However, due to the reviewer’s comment we found that we did not properly name the subgenotype in figure 5c and fixed that in the revised manuscript.
It looks as if one panel is missing in the figure1. Indeed, the figure 1c is not related to HEV RNA but Ag (line 164) and the figure 1e does not exist (line 172).
Response: In an earlier version of our manuscript we presented data of figure 1a in two separate panels 1a (HEV RNA) and 1b (HEV ORF2 Ag). However, we found that one panel for both RNA and Ag would be sufficient but clearly missed to edit the text accordingly. We thank the reviewer for highlighting this mistake, which has been corrected in the revision (lines 178, 180, 184 and 192).
The name of the media must be added in the figure 3b to help the reader.
Response: We apologize for this inaccuracy. We added the information into the revised figure.
Please explain why the strain 3e was not tested on HepG2C3A (figure 3c).
Response: The gt 3e strain was tested on HepG2/C3A but the cell line did not support the replication even when the cells were cultivated in optimized medium. That’s why all three dashed lines of the radar plot drop to “n.d.” (not detected). The same was observed with HuH-7-Lunet BLR cultivated in non-optimized BMEM and inoculated with gt 3e strain – the red dashed line drops to “n.d.”. To avoid any confusion, we tried to emphasize this circumstance clearer in the text (lines 265-267).
Figure 5, last left panel (gt3f) : it is surprising the 10 -4 dilution grows while the 10-3 does not.
Response: We can only speculate about this finding which may be due to one or a combination of three reasons:
(1) The PCR’s limit of detection (95% LoD) which is at 1,000 copies/mL. The detected viral load of 729 copies/mL may therefore be missed if retested. The detection probability at this concentration is around 30–60%.
(2) Time of observation: If cells would have been observed for a longer time, the 10 -4 dilution might have started growing at 28 days post inoculation.
(3) A stochastiv effect.
However, as this is a single observation, no extensive discussion has been included into the manuscript.

Reviewer 2 Report
A very thorough study of HEV propagation in cell culture which will be of great use and interest to many scientists in this field. The findings have been well presented but I do have some concerns listed below which are mostly quite minor. I enjoyed reading this paper.
Intro
ORFs 1, 2 and 3 are described but the putative ORF4 of certain genotypes is not mentioned.
Methods
In the inocula section only serum is described. However, faecal and plasma inoculum are used in the study. These should be in the methods section with details of source (same as sera?).
Also the HEV viral loads of all the clinical isolates should be given in the methods.
Were plasma or serum samples treated for envelope removal prior to use as an inoculum?
How were the faecal samples used- was a suspension made? If so how was this made?
The study also uses the culture derived strain 47832c so details of how this was prepared should be in the methods- what cell type was it propagated in and with what media type prior to use? This is important as it may be adapted to these culture conditions and could skew the results where cell type and culture conditions are trialled.
What was the viral load of the 47832c stock?
Results
With some of the data presented the number of replicates is given in the legend, but not for all. I think it is important for the reader to have this information for all the data presented.
Also, when the experiments were carried out in duplicate or more, the mean value is presented but no error bars are included, would the author consider including these?
In figure 1C, I think it is worth noting that the faecal derived 14-16078 demonstrated an increased antigen production in cells that were washed 5x/week, as this is contrary to the findings for 47832C, where 2x/week washes were acceptable and the finding that is stated in the text.
In section 3.2, I think it would help to explain how the seeded 0 days prior to inoculation was performed, were the cells given time to adhere to the flask prior to inoculation?
On pg 5, line 198 I would suggest changing ‘earlier’ to ‘at least’. I think stating ‘earlier’ suggests that there is a continuously proportional relationship with time. However, seeding earlier than 10 or 7 days prior to infection did not yield higher viral loads.
I disagree with the statement on lines 202-203. For A549 cells there is little difference when comparing the different cell densities. For PLC/PRF, there was only a difference when comparing the very lowest cell density at a single time prior to inoculation (day -10).
I found figure 3b hard to understand and wondered if perhaps some data labels are missing?
I would suggest changing line 241 ‘MEMM showed slightly better results…..’ to ‘MEMM showed better results in 3 out of 4 cell lines…’
In figure 4c, the results are quite different from 4a and 4b, particularly for the cell line HepG2/C3A, despite there only being a small increase in the viral load in the inoculation, do you know why this may be?
In regards to section 3.6- is it known what proportion of cells were infected? Was it 100%? Although no CPE was detected, was the proliferation rate of infected cells measured? If so, was there any difference between infected and non-infected cells?
I figure 6 ‘splitted’ should be ‘split’
In figure 6b there is only data from one cell line where as the others have several? Was this carried out for other cell lines?
Discussion
Can the authors offer any opinion on why over confluent /3d layers improve infection rates? Do they believe it is due to improved cell to cell transmission, or perhaps changes to the cytoskeleton, or something else?
I wonder if the authors might want to touch on why the genotypes studied here proliferate more efficiently in cell culture than types 1 and 2?
Obviously the cell lines which support HEV culture best, and are tested here, are all continuous immortalised lines, which many not be reflective of cells in vivo. I think it is important to comment on their potential limitations in this regard.
Also, the authors might want to highlight that the cell lines used in this study were all of human origin and that the establishment of culture systems using animal derived cells may be useful in studying the zoonotic ability of HEV3, HEV4 and HEV7?
Author Response
Response to Reviewer 2 Comments
A very thorough study of HEV propagation in cell culture which will be of great use and interest to many scientists in this field. The findings have been well presented but I do have some concerns listed below which are mostly quite minor. I enjoyed reading this paper.
Response: We thank the reviewer for this positive judgement on our study.
Intro
ORFs 1, 2 and 3 are described but the putative ORF4 of certain genotypes is not mentioned.
Response: We thank the reviewer for pointing out the lack of ORF4 description. We included that ORF4 –which is to our knowledge exclusively expressed in genotype 1– controls the activity of the RNA dependent RNA polymerase of HEV (lines 45-46).
Methods
In the inocula section only serum is described. However, faecal and plasma inoculum are used in the study. These should be in the methods section with details of source (same as sera?).
Response: This is right, we focussed our description only on the three materials which we successfully isolated HEV from. However, we agree that the other materials should also be mentioned. We added the missing information to the section (lines 108-111).
Also the HEV viral loads of all the clinical isolates should be given in the methods.
Response: We agree and added information on viral loads to the respective materials section (lines 106-111).
Were plasma or serum samples treated for envelope removal prior to use as an inoculum?
Response: We thank the reviewer for bringing up this point, which was also raised by reviewer 1. In order to improve the clarity of the text, we revised the description of the inoculum preparation and added a sentence explaining that we did nothing beyond that (line 121).
How were the faecal samples used- was a suspension made? If so how was this made?
Response: We used one faecal sample in the study from which we prepared a 10% faecal suspension in PBS without Ca2+ and Mg2+ containing 0.2% BSA (w/v). We added these details to the inocula section (lines 118-121).
The study also uses the culture derived strain 47832c so details of how this was prepared should be in the methods- what cell type was it propagated in and with what media type prior to use? This is important as it may be adapted to these culture conditions and could skew the results where cell type and culture conditions are trialled.
Response: We thank the reviewer for mentioning this important aspect of the study. We used HEV 47832c positive supernatant as a positive control in every optimization experiment –as well as all novel isolates available at that time– and therefore freshly prepared it for each experiment (lines 115-116). Moreover, after every optimization experiment the culture conditions were adjusted to the best fitting parameters. Hence, we used HEV 47832c positive supernatant from A549 and PLC/PRF/5 cultured under differing conditions. We therefore specified each “culture supernatant” in the results section to the type of cell culture supernatant.
Besides that, we always used a patient material in optimization experiments to uncover possible cell culture adaption bias.
What was the viral load of the 47832c stock?
Response: As mentioned before, viral stocks were always freshly prepared since they served as positive controls. Therefore, the respective viral load is always stated in the results section individually in each subitem of an optimization experiment.
Results
With some of the data presented the number of replicates is given in the legend, but not for all. I think it is important for the reader to have this information for all the data presented.
Response: Due to technical reasons (e.g. the maximum amount of cell culture flasks managable at a time) not all experiments were performed as two or more replicates. We therefore only mentioned replicates as long as an experiment covered these.
Also, when the experiments were carried out in duplicate or more, the mean value is presented but no error bars are included, would the author consider including these?
Response: We had discussed if standard deviations should be displayed as error bars in the graphs. We concluded that some graphs (figure 2a, 3b, 4a & 4b) would benefit from error bars whereas others would become rather confusing (figure 3c & 4c) or give a wrong impression (figure 2b) in our opinion. Therefore, we chose to not display any error bars to keep the overall presentation consistent. We would like to explain in more detail why we worry about wrong impressions and possibly causing confusion:
The data of figure 2b consists of technical duplicates. In four cases (3 × 105 cells 7 days prior to cell seeding (dpcs), 1 × 106 cells 7 dpcs, 3 × 106 cells 3 dpcs and 9 × 106 cells 0 dpcs), HEV replication at 28 days post inoculation was not detectable in one duplicate, whereas the other duplicate had detectable levels of HEV RNA. The mean of a fictitious viral load of 104 HEV RNA copies/mL and a second, non-detectable load would still be 104 c/mL. The standard deviation would be zero. Therefore, the data would be interpreted as a robust infection although HEV replication did only happen in one of two cases. We therefore chose to display the data of duplicates singly instead of mean and error bars.
The radar plots contain data points expressed as lines of three (figure 3c) and four (figure 4c) investigated parameters. In some cases, the lines cluster close together and error bars made the data really hard to read and to distinguish from each other. For the sake of clarity, we therefore resigned from using error bars.
In figure 1C, I think it is worth noting that the faecal derived 14-16078 demonstrated an increased antigen production in cells that were washed 5x/week, as this is contrary to the findings for 47832C, where 2x/week washes were acceptable and the finding that is stated in the text.
Response: We agree with this comment and added the observation of slightly elevated HEV ORF2 antigen levels in supernatant of A549 inoculated with the faecal suspension and refreshed five times a week (lines 192-194). However, the detected antigen levels did not culminate in detectable replication 7 weeks after inoculation.
In section 3.2, I think it would help to explain how the seeded 0 days prior to inoculation was performed, were the cells given time to adhere to the flask prior to inoculation?
Response: We thank the reviewer for pointing us this missing aspect. Cell suspensions were adjusted to the defined concentrations and transferred to a T12.5 in a total volume of 2.5 mL. Immediately after cell seeding, 250 µl of inoculum was added while cells were still in suspension. This information was added to section 3.2 (lines 216-218).
On pg 5, line 198 I would suggest changing ‘earlier’ to ‘at least’. I think stating ‘earlier’ suggests that there is a continuously proportional relationship with time. However, seeding earlier than 10 or 7 days prior to infection did not yield higher viral loads.
Response: This point is very well taken. A549 do not generate higher viral loads at 14 dpi when cells were cultivated longer than 7-10 days. We replaced “earlier” by “at least 7-10 days (depending on cell concentration)” as suggested by the reviewer (lines 220-221).
I disagree with the statement on lines 202-203. For A549 cells there is little difference when comparing the different cell densities. For PLC/PRF, there was only a difference when comparing the very lowest cell density at a single time prior to inoculation (day -10).
Response: We agree with the reviewer that the differences are not huge but a clear tendency cannot be denied. We would also like to point the reviewer to time point “-14” at which 9 × 106 viable PLC/PRF/5 seeded produce at least 3 to 20 times higher viral loads at 28 dpi compared to cells seeded at less dense concentrations. Moreover, the infection of PLC/PRF/5 is more reliable when taking a look at time point “-7”. Both duplicates are only infected when cells are seeded at the highest concentration. We now modified the statement to “Generally, we observed a tendency to more reliable infection and higher viral loads when...” (lines 227-229).
I found figure 3b hard to understand and wondered if perhaps some data labels are missing?
Response: We apologise for the missing labels on the x axis of figure 3b. We replaced the figure with the complete version.
I would suggest changing line 241 ‘MEMM showed slightly better results…..’ to ‘MEMM showed better results in 3 out of 4 cell lines…’
Response: We thank the reviewer for describing the situation more precisely and adapted the wording accordingly (line 268-269).
In figure 4c, the results are quite different from 4a and 4b, particularly for the cell line HepG2/C3A, despite there only being a small increase in the viral load in the inoculation, do you know why this may be?
Response: We can only speculate on the underlying reason. An adaption of isolates to cell culture cannot be completely excluded. Cell lines in figure 4c were inoculated with isolates being passaged once more than isolates used to inoculate cells in figure 4a and b.
In regards to section 3.6- is it known what proportion of cells were infected? Was it 100%? Although no CPE was detected, was the proliferation rate of infected cells measured? If so, was there any difference between infected and non-infected cells?
Response: We regret that we cannot provide adequate answers to these questions.
Long-term cultivation of HEV positive cells mostly relied on medium refreshment rather than splitting and therefore only results from supernatants tested by HEV PCR are available. Long-term cultured cells themselves were never systematically investigated e.g. by immunofluorescence staining. The cell proliferation rate was not tested but this hint is very valuable and will be definitely implemented in future experiment designs.
I figure 6 ‘splitted’ should be ‘split’
Response: We apologise for this mistake and corrected the captions.
In figure 6b there is only data from one cell line where as the others have several? Was this carried out for other cell lines?
Response: In addition to PLC/PRF/5, this was also carried out for A549, HepG2/C3A, HuH-7-Lunet BLR and MRC-5. However, the latter cell lines were followed up for 11 weeks which is only ~10% of the follow-up time of PLC/PRF/5. In this context, we did not classifiy 11 weeks as long-term cultivation and therefore did not include the data into figure 6b.
Discussion
Can the authors offer any opinion on why over confluent /3d layers improve infection rates? Do they believe it is due to improved cell to cell transmission, or perhaps changes to the cytoskeleton, or something else?
Response: This is a good question. The reason for improved infection rates in 3d layers could be due to several points but we can only speculate on that point. As mentioned by the reviewer, it could be due to improved cell to cell transmission, possibly by bringing more cells into direct contact to each other. It could also be due to cell proliferation and differentiation enabling more sufficient HEV infection. Autophagy may also play a role in this context. We would like to catch up on this in future projects and added an explanatory passage to the discussion (lines 434-438).
I wonder if the authors might want to touch on why the genotypes studied here proliferate more efficiently in cell culture than types 1 and 2?
Response: We would rather prefer not to claim that our isolates proliferate more efficiently in cell culture than isolates of genotypes (gt) 1 and 2. A recent work by Wu and Dao Thi et al. 2018 (Gastroenterology) showed that strains Sar-55 (gt 1), Mexico-14 (gt 2) and US-2 (gt 3) replicate to comparable viral loads of ~106 HEV RNA copies/ µg total RNA at 7 days post infection in hepatocyte-like cells derived from human embryonic and induced pluripotent stem cells. Solely strain TW6196E (gt 4) replicated to lower viral loads of ~2 × 105 HEV RNA copies/ µg total RNA.
Unfortunately, we don’t have access to HEV other than gt 3 but would happily take any material positive for HEV genotypes other than 3 and isolate these strains in our optimized cell culture system to investigate such questions in the future.
Obviously the cell lines which support HEV culture best, and are tested here, are all continuous immortalised lines, which many not be reflective of cells in vivo. I think it is important to comment on their potential limitations in this regard.
Response: This is an important point which we must have missed while mainly focussing on establishing a system susceptible to clinically prevalent strains. We added this limitation as suggested by the reviewer (lines 484-485).
Also, the authors might want to highlight that the cell lines used in this study were all of human origin and that the establishment of culture systems using animal derived cells may be useful in studying the zoonotic ability of HEV3, HEV4 and HEV7?
Response: We completely agree. Studying the zoonotic features of HEV is certainly an important aspect, which has been included now (lines 486-487). A recent review on HEV which we cited in our manuscript (Meister et al. 2019, Antiviral Research) sums up quite nicely in Table 3 that human derived cell lines (for instance, PLC/PRF/5 amongst others) have the potential to support replication of animal derived HEV3, HEV4 and HEV7. This was also investigated vice versa, albeit to a lesser extent: Shukla et al. (2011, PNAS) showed that HEV of human origin can replicate in swine and deer cells.

Reviewer 3 Report
The authors present a novel cell culture system for HEV. They optimized conditions, such as medium composition, cell density and cell type. They used the optimized system to recover and cultivate clinical isolates of genotype 3 HEV. Identification of optimal parameters for the cell culture system has been done thoroughly and extensively. Isolation of clinical isolates and their cultivation is of great interest. The impact of the paper could be improved by the following adjustments and recommendations:
Line 64 and line 373: The authors claim that the effort in maintenance was significantly reduced. How much is the effort reduced? Is that an important feature of their method?
Lines 380-384: It is appreciated that the authors give a summary of the optimized cell culture procedure, but it would be better to move this short paragraph to an appropriate point in the results section.
Lines 60-62: in the introduction section, the authors claim that improved cell culture systems could be used to “generate a classic inactive or attenuated vaccine, to develop and test specific drugs or to identify strategies for efficient HEV inactivation.” It would be of great relevance to demonstrate that their cell culture system is suitable for that by testing Ribavirin against 47832c as well as the clinical isolates.
Introduction: The authors should mention the HEV gt3 strain 83.2 by the Wakita lab that replicates in cell culture and is used extensively for example by Gouttenoire et al. Plos pathogens 2018. Further, the S17 insertion in the p6 strain, which is the most commonly used isolate, is mentioned as uncommon or usual. In the discussion, it is mentioned that this insertion was present in the original feces of the patient (Shukla et al. PNAS 2011), just as a minor species and was selected during cell culture passage. To avoid confusion, this should be mentioned in the introduction section. Citation Shukla et al: “RT-PCR with paired HEV and insertion sequence primers detected viral genomes with the insertion in the original fecal suspension, indicating that a double-recombination event had occurred either in the patient or in a previous host”.
Figure 2:
It seems peculiar that there is only one condition (3*105 cells PLC/PRF/5 cells seeded around 35 days before infection), where replication exceeds the dose used for infection, although many graphs recovery of virus after initial decrease. Please comment on that fact and add the paragraph in the text?
Figure 3:
Figure 3A+ 3B: three independent experiments and statistics should be performed.
Figure 5: For the sake of visibility, it is advised to include the TCID50 values as some sort of bar or graph into figure 5.
Author Response
Response to Reviewer 3 Comments
The authors present a novel cell culture system for HEV. They optimized conditions, such as medium composition, cell density and cell type. They used the optimized system to recover and cultivate clinical isolates of genotype 3 HEV. Identification of optimal parameters for the cell culture system has been done thoroughly and extensively. Isolation of clinical isolates and their cultivation is of great interest. The impact of the paper could be improved by the following adjustments and recommendations:
Response: We thank the reviewer for this constructively positive overall appraisal of our study.
Line 64 and line 373: The authors claim that the effort in maintenance was significantly reduced. How much is the effort reduced? Is that an important feature of their method?
Response: We missed to clearly state that some HEV cell culture systems rely on daily refreshment of half of the medium. We found that twice a week completely is absolutely sufficient. Therefore, the effort is reduced by ~70%. We wouldn’t call it a feature but see it as an aspect to facilitate the work on HEV cultivation. We thank the reviewer for bringing this up and added information on daily hands-on time to both lines (lines 73 and 414).
Lines 380-384: It is appreciated that the authors give a summary of the optimized cell culture procedure, but it would be better to move this short paragraph to an appropriate point in the results section.
Response: We agree and introduced a new subitem into the results section entitled “Optimized Method for Successful HEV Isolation from Serum” (lines 390-400). In addition, we extended the summary with details about cell cultivation temperature as well as sample preparation which was issued by reviewer 1 and 2.
Lines 60-62: in the introduction section, the authors claim that improved cell culture systems could be used to “generate a classic inactive or attenuated vaccine, to develop and test specific drugs or to identify strategies for efficient HEV inactivation.” It would be of great relevance to demonstrate that their cell culture system is suitable for that by testing Ribavirin against 47832c as well as the clinical isolates.
Response: We agree with the reviewer’s suggestion for testing Ribavirin against all isolates in our optimized cell culture system. However, we regret this is out of scope of the current manuscript and we are planning investigations on this topic in future projects.
Introduction: The authors should mention the HEV gt3 strain 83.2 by the Wakita lab that replicates in cell culture and is used extensively for example by Gouttenoire et al. Plos pathogens 2018. Further, the S17 insertion in the p6 strain, which is the most commonly used isolate, is mentioned as uncommon or usual. In the discussion, it is mentioned that this insertion was present in the original feces of the patient (Shukla et al. PNAS 2011), just as a minor species and was selected during cell culture passage. To avoid confusion, this should be mentioned in the introduction section. Citation Shukla et al: “RT-PCR with paired HEV and insertion sequence primers detected viral genomes with the insertion in the original fecal suspension, indicating that a double-recombination event had occurred either in the patient or in a previous host”.
Response: Although we quoted the work of Gouttenoire et al., we missed to clearly mention HEV 83.2 strain in the text, which is now included (lines 63-64). In addition, we introduced a new reference which describes the generation of the infectious clone (Shiota et al. (2013) JVI) (line 63).
In agreement with the reviewer’s suggestion we added the following sentence to the introduction: “The insertions are thought to be aquired by recombination events either in the patient or in a previous host” (lines 58-60).
Figure 2: It seems peculiar that there is only one condition (3*105 cells PLC/PRF/5 cells seeded around 35 days before infection), where replication exceeds the dose used for infection, although many graphs recovery of virus after initial decrease. Please comment on that fact and add the paragraph in the text?
Response: We laboriously learned with each additional experiment that HEV is a very slow replicating virus. The maximum viral load may be reached after 4 weeks post inoculation (wpi) but can also take up to 10 wpi. Therefore, it is not unusual for HEV –as shown in figure 2– that testing of the last supernatant collection day (28 dpi) did not result in higher or comparable viral loads as the inoculum. To prevent any confusions of the reader about this aspect, we added a short explanatory sentence to section 3.2 (lines 225-226).
Figure 3: Figure 3A+ 3B: three independent experiments and statistics should be performed.
Response: We agree that triplicates would make the conclusions even stronger with respect to medium supplements. However, the described experiments were conducted only once. Unfortunately, we cannot adequately address this issue within the revision timeframe (10 days). The experiment alone would take 9 weeks in total (2 weeks to expand the cells, 2 weeks to let cells grow overconfluently, 4 weeks to reach time point 28 dpi, 1 week to test supernatants by ELISA and PCR and analyse the data). A third independent experiment could start 2 weeks later, leading to ~3 months to insert additional data to the revised manuscript. Figure 3C already confirms the positive effect of medium supplements (shown in figures 3A and B) on HEV replication and infection. We hope for the reviewer’s understanding that we think the high efforts of additional experiments are not adequate in relation to the moderate improvement of the data.
Figure 5: For the sake of visibility, it is advised to include the TCID50 values as some sort of bar or graph into figure 5.
Response: We agree with this reviewer’s advice. We included dashed lines representing the respective TCID50 values as a guide for the reader in figure 5. We also added an additional sentence to the figure legend, which explains that these values were determined in a subsequent experiment at day 13 post inoculation.

Round 2
Reviewer 1 Report
I thank the authors for answering all my concerns.
Reviewer 2 Report
Thank you to the authors for address the points I noted. I am happy with the responses and changes that were implemented.
Reviewer 3 Report
Some of the issues could not be addressed due to time reseasons. Otherwise all points addressed.